# Effect of Combined Infrared and Hot Air Drying Strategies on the Quality of Chrysanthemum (*Chrysanthemum morifolium* Ramat.) Cakes: Drying Behavior, Aroma Profiles and Phenolic Compounds

**DOI:** 10.3390/foods11152240

**Published:** 2022-07-27

**Authors:** Huihuang Xu, Min Wu, Yong Wang, Wenguang Wei, Dongyu Sun, Dong Li, Zhian Zheng, Fei Gao

**Affiliations:** 1College of Engineering, China Agricultural University, No. 17 Qinghua East Road, Haidian District, Beijing 100083, China; xhhhenan@163.com (H.X.); wwg5946@163.com (W.W.); sdymavis@163.com (D.S.); dongli@cau.edu.cn (D.L.); zhengza@cau.edu.cn (Z.Z.); 2School of Chemical Engineering, University of New South Wales, Sydney, NSW 2052, Australia; benjaminwy@gmail.com; 3School of Food and Health, Beijing Technology and Business University, Haidian District, Beijing 100083, China; sophia.gophi@gmail.com

**Keywords:** medicinal chrysanthemum, combined infrared and hot air drying, drying behavior, aroma profiles, phenolic compounds, microstructure

## Abstract

Chrysanthemum (*Chrysanthemum morifolium* Ramat.) is a seasonal plant with high medicinal and aesthetic value, and drying is an effective practice to enhance its storability after harvesting. The effects of hot air drying (HAD), combined infrared and hot air drying (IR-HAD), and sequential IR-HAD and HAD (IR-HAD + HAD) on the drying behavior, color, shrinkage, aroma profiles, phenolic compounds, and microstructure of chrysanthemum cakes were studied. Results showed that the increasing temperature resulted in a decrease in drying time and an increase in drying rate and moisture diffusivity. The Logarithmic and Page models exhibited superior fit in describing the dehydration process. Among the three drying strategies, IR-HAD was more effective in reducing energy consumption, improving shrinkage, water holding capacity, water binding capacity and cellular microstructure, while IR-HAD + HAD showed better inhibitory effect on color deterioration. Furthermore, gas chromatography–mass spectrometry (GC-MS) analysis revealed that different drying strategies dramatically influenced the aroma profiles in samples, and IR-HAD obtained the highest concentration of volatiles. The results of ultra-performance liquid chromatography (UPLC) indicated that the introduction of infrared radiation contributed to increasing the contents of chlorogenic acid, luteolin, total phenolic and flavonoid. These suggested that IR-HAD was a promising technique for drying medicinal chrysanthemum.

## 1. Introduction

Chrysanthemum (*Chrysanthemum morifolium* Ramat.) is a traditional medicinal plant commercially cultivated in oriental countries for its medicinal benefits and aesthetic value [1]. The most precious part of chrysanthemum, the capitulum (flower), is used as raw material for Chinese medicine and health teas because of its therapeutic potential and unique flavor [2]. Current research has reported the predominant bioactive components in chrysanthemum, mainly flavonoids, phenolic acid, and triterpenoids. Thereinto, chlorogenic acid and luteolin are the important phenolics with extensive biological capacities such anti-oxidant and anti-inflammatory functions, as well as hypolipidemic abilities [3]. Apart from medicinal properties, chrysanthemum has drawn increasing attention in the cosmetology industry because of its aromatic properties [4]. Petals are the most ornamental part of chrysanthemum as they are not only an essential origin of fragrance, but their color and shape also affect consumer preferences. However, season-specific and perishability limit the year-round availability of fresh chrysanthemums. Therefore, drying is commonly used as an effective practice to prolong shelf life and develop chrysanthemum-based products such as loose tea or tea cake.

Hot air drying (HAD) is widely used in industrial production owing to its simple equipment and broad applicability [5]. However, prolonged exposure to aerobic environment and high temperatures may result in the loss of bioactive and volatile ingredients [6]. Additionally, undesired changes in the appearance of chrysanthemum may occur, such as browning and curling of petals. Freeze-drying (FD) offers advantages in terms of improving the bioactive composition and physical structure the product [7]. Unfortunately, it was not economically realistic to dry seasonal herbs such as chrysanthemum due to high energy consumption and investment. Infrared (IR) heating is an emerging technology for rapid drying of thin-layered biomaterials. Previous studies have shown that the levels of total phenols and total flavonoids in IR-dried chrysanthemums were significantly (*p* < 0.05) higher than in shade-dried ones [8]. However, the radiation energy decays gradually as the penetration depth increases, which limits the effectiveness of IR in deep bed drying [9].

To compensate for the shortcomings of individual technology, IR is generally combined with other drying technologies. Our previous studies have demonstrated that combined infrared and hot air drying (IR-HAD) has a higher drying efficiency and a better inhibitory effect on chrysanthemum color deterioration than HAD alone [10]. During IR-HAD, high-speed air directly impacted the material surface, forming a thin boundary layer on the material surface, enhancing the heat and mass transfer efficiency. The improved drying efficiency implies that the drying time is dramatically shortened, which will help reduce the loss of heat-labile biomacromolecules [11]. Notably, infrared radiation has been reported to possess a more potent cell-damaging effect compared to HAD and FD. This may trigger chemical reactions that favor the enrichment of aroma profiles in plants [12]. Currently, infrared-assisted drying techniques have been successfully applied to the process of rose flower [13], *Bletilla striata* (Thunb.) Reichb.f. flower [14], agarwood leaves [15] and other edible plants. However, in some application scenarios, individual drying techniques with a single heating mechanism may not be sufficient in satisfying the requirements for product quality, process stability and safety. Therefore, multiple individual drying technologies with different heating mechanisms could be applied successively or intermittently to improve drying efficiency, and product quality, such as sequential air jet impingement and hot air-assisted radio frequency drying [16] and two-stage sequential microwave and hot air drying [17]. Although there were numerous studies regarding HAD and IR, these studies have mainly been performed individually with no evolution of the combination of infrared and hot air drying of chrysanthemum, especially the combination strategies and process-quality interaction.

The metabolic activity in chrysanthemum was highly susceptible to the temperature during drying, which accordingly leads to the accumulation or loss of phenolic and volatile compounds. For example, non-enzymatic reactions have been reported to be a potential pathway for chlorogenic acid production [18]. In addition, a high temperature may result in evaporative losses of highly volatile aroma components, such as terpenes. It is worth noting that the aroma of chrysanthemum is an extremely complex matrix that includes volatile compounds of multiple chemical classes such as terpenes, alcohols, aldehydes, ketones and phenols. The retention levels of these compounds are related to physical evaporation, chemical degradation and dynamic chemical transformations between different compounds during drying [19], which would have a critical impact on the sensory quality and acceptability of the final product. When the number and type of variables are very large and the data set is complex, chemometrics is often employed, using various statistical and mathematical algorithms to maximize the relevant chemical information [20]. Principal component analysis (PCA) and hierarchical cluster analysis (HCA) are commonly used chemometric tools to identify and classify the similarities/differences between variables [21]. Farag et al. [22] successfully achieved rapid classification of essential oils based on their chemical composition using PCA and HCA. Although PCA and HCA allow for a clear view of the association between variables, they lack a measure to the association. Pearson correlation analysis offers an alternative solution to this problem, which allows for the quantitative measurement of association between variables by calculating correlation coefficients [23]. However, there is a lack of studies focusing on assessing the association between the aroma components of dried chrysanthemums and drying methods. Therefore, based on the similarities/differences in the retention levels of volatile compounds, chemometric tools can be employed to differentiate between chrysanthemum cakes dried by different methods and to quantitatively analyze the association between drying characteristics and aroma profile of dried chrysanthemum cakes.

This study aimed to investigate the effects of different combined infrared and hot air-drying strategies on the drying behavior, physicochemical properties and microstructure of medical chrysanthemum-based products. The details include: (1) the study of the drying behavior of chrysanthemum cakes under different conditions; (2) examination of the color, shrinkage and aroma profiles under different drying conditions; (3) investigation of the chlorogenic acid, luteolin, total phenolic content, total flavonoid content, water holding capacity, water binding capacity and microstructure of chrysanthemum cakes under different drying conditions. The results will have far-reaching implications for the application of infrared drying technology in the industrial production of dried chrysanthemums.

## 2. Materials and Methods

### 2.1. Materials

Chrysanthemum (*Chrysanthemum morifolium* Ramat.) were collected from the experimental field of China Agricultural University (Beijing, China). Fresh flowers with uniform physical characteristics were manually selected and laid evenly in a steamer with a diameter of 11 cm and a depth of 4 cm. Prior to the experiments, the samples were steamed into a cake shape using a superheated steam impingement blanching equipment. The average weight and thickness of chrysanthemum cakes were 78.48 ± 1.40 g and 17.35 ± 1.01 mm, respectively. The average moisture content (wet basis) of steamed samples was 75.43 ± 2.96%, which was determined by oven-drying at 105 °C for 24 h [24].

### 2.2. Drying of Chrysanthemum Cakes

#### 2.2.1. Freeze-Drying (FD)

Freeze-dried chrysanthemum cakes were used as the control group. Samples were rapidly frozen and cooled to below 30 °C, then placed in a vacuum chamber and lyophilized using a freeze dryer (LGJ-18C, Beijing Sihuan Scientific Instrument Factory Co., Ltd., Beijing, China) under a vacuum of 20 Pa, a condenser temperature of −60 °C and a tray temperature of 25 °C.

#### 2.2.2. Hot Air Drying (HAD)

A 1.8 kW hot air dryer used in this study was developed independently by the research team (Figure 1A,B) and was designed with dimensions of 980 mm × 400 mm × 960 mm, mainly including a heating system, automatic temperature regulation system, and electronic control system [25]. The capacity of the dryer is 35 × 35 × 30 cm. Before the start of each experiment, the drying temperature (55, 65 and 75 °C) and air speed (2 m/s) were set by touch screen, and the dryer was preheated for 30 min to ensure the stable temperature in the drying chamber. The chrysanthemum cakes with the weight of 78.48 ± 1.40 g and the thickness of 17.35 ± 1.01 mm were put into the drying chamber, and the mass loss was recorder by weighing them with electronic balance (SP402, Ohaus Co., Parsippany, NJ, USA) during drying. An electric meter was installed at the output of the dryer to evaluate the energy consumption of each drying experiment.

#### 2.2.3. Combined Infrared and Hot Air Drying (IR-HAD)

A laboratory-scale combined infrared and hot air dryer jointly developed by our research team and STC Taizhou Senttech Infrared Technology Co., Ltd. (Taizhou, China) was used. The composition of the equipment is shown in Figure 1C,D, which mainly consisted of the drying chamber, infrared tubes, centrifugal fan, and control panel [26]. The volume of the dryer is 26 × 35 × 40 cm. The heating sources are six infrared lamps with wavelength bands of 0.75–4 μm. Chrysanthemum cakes were placed flat on a material tray and dried at an air velocity of 2.0 m/s, a radiation distance of 12 cm, and heating temperatures of 55, 65 and 75 °C, respectively. Similarly, mass loss and power consumption were recorded in the same way as described in Section 2.2.2.

#### 2.2.4. The Sequential Drying (IR-HAD + HAD)

During the sequential drying experiments (IR-HAD + HAD), the prepared chrysanthemum cakes were first placed in the drying chamber of the IR-HAD apparatus, and dried at set temperature (55, 65 and 75 °C), air speed (2.0 m/s) and radiation distance (12 cm). Based on our preliminary experiments, the IR-HAD drying time was chosen to be 180 min, which reduced the moisture content of the sample to approximately 30 g/100 g. After 180 min of IR-HAD, the samples were quickly transferred to the HAD apparatus using the same drying temperature and air speed parameters as in the first stage until the end of drying.

### 2.3. Drying Characteristics

#### 2.3.1. Moisture Ratio (*MR*) and Drying Rate (*DR*)

The moisture ratio (*MR*) of chrysanthemum cakes under different drying conditions was determined using Equation (1).
(1)$$MR=\frac{M_{t}-M_{e}}{M_{o}-M_{e}}$$
where *M*_0_, *M_t_* and *M_e_*, indicate the initial moisture content, moisture content at time *t* and equilibrium moisture content (g water/g dry basis), respectively.

The drying rate (*DR*) was employed to describe the speed of moisture removal during drying and was determined using Equation (2).
(2)$$DR=\frac{M_{to}-M_{t1}}{t_{1}-t_{0}}$$
where *M_t_*_1_ and *M_t_*_0_ indicate the dry-based moisture content (g/(g·min)) at times *t*_1_ and *t*_0_, respectively, and *t*_0_ < *t*_1_ (min).

#### 2.3.2. Effective Moisture Diffusivity (*D_eff_*)

Effective moisture diffusivity (*D_eff_*) was estimated using Fick’s second law of diffusion. The chrysanthemum cake was assumed to be a uniform, infinite slab, thus its thickness could be regarded as the migration distance of moisture. For a long drying time, the analytical solution of Fick’s second diffusion equation is expressed by Equation (3).
(3)$$MR=\frac{M_{t}}{M_{o}}\approx \frac{8}{\pi^{2}}\exp(\frac{\pi^{2}D_{eff}}{H^{2}}t)$$
where *H* and *t* represent the thickness of the chrysanthemum cake (*m*) and the drying time (s), respectively.

Equation (3) was converted into its logarithmic form (Equation (4)), the gradient (*k*) was calculated by performing a linear regression of the ln (*MR*) versus time curves, and *D_eff_* was then obtained using Equation (5).
(4)$$\ln(MR)=\ln\frac{8}{\pi^{2}}-\frac{\pi^{2}D_{eff}}{H^{2}}t$$
(5)$$D_{eff}=-\frac{H^{{}^{2}}}{\pi^{2}}k$$

#### 2.3.3. Activation Energy (*E_a_*)

The activation energy (*E_a_*) was described by the Arrhenius-type equation.
(6)$$D_{eff}=D_{o}\exp\frac{E_{{}_{a}}}{R(T+273.15)}$$

Taking the natural logarithm of Equation (6) to obtain Equation (7),
(7)$$\ln(D_{eff})=\ln(D_{o})-\frac{E_{a}}{RT}$$
where *D_o_* is the diffusivity basis (m^2^/s), *R* is the universal gas constant (kJ/mol), *T* is the drying temperature (°C). The *E_a_* can be determined by calculating the slope of the plot of ln(*D_eff_*) versus 1/(*T* + 273.15).

### 2.4. Mathematical Modeling

The drying process of chrysanthemum cakes under different drying conditions is described using four mathematical models (Table 1). The applicability of the models was evaluated based on the model constants (*k*, *a*, *b*, *t*, and *n*) and model selection criteria (*R*^2^, *RMSE*, *SSE*, *AIC* and *BIC*) obtained from MATLAB software [27]. Higher *R*^2^ (coefficient of determination) values represent a stronger correlation, and lower *RMSE* (root mean square error) and *SSE* (sum squared error) values indicate higher reliability of the model.

### 2.5. Specific Energy Consumption

Energy consumption is an important indicator for assessing the performance of three drying methods. Therefore, based on the power consumption of the dryer under different drying conditions measured by an electric meter, the energy consumed to remove 1 kilogram of water from the chrysanthemum cake was estimated using Equation (8), also called specific energy consumption (*SEC*) (kW h/kg).
(8)$$SEC=\frac{1000E}{M_{o}X_{o}-M_{i}X_{i}}$$
where *M_o_* and *M_i_* indicate the mass of samples at the beginning and end of drying (g), respectively; *X_o_* and *X_i_* indicate the moisture content of samples at the beginning and end of drying (w.b., g/g), respectively; *E* indicates the electrical energy consumed for drying (kW h).

### 2.6. Appearance of Chrysanthemum Cakes

#### 2.6.1. Shrinkage Ratio

A computer vision system (CVS) equipped with an industrial camera (Basler AG, Hamburg, Germany) was employed to capture chrysanthemum images (Figure 2). Subsequently, the information of captured images was obtained using the image processing method [28]. The applied lens was a fixed-focus lens focused on the background plate during the image acquisition. Therefore, the total number of pixels in the white region of the binary image obtained by image segmentation was equivalent to the sum of the pixel number of chrysanthemum cakes. The process of calculating the areas of chrysanthemum cakes mainly includes the following steps: conversion to grayscale, generation of binary images, erosion and expansion operations, hole-filling operations, and calculation of the number of pixels.

The shrinkage ratio (*SR*) was calculated by the following Equation (9),
(9)$$SR=\frac{A_{o}-A_{1}}{A_{o}}$$
where *S* refers to the shrinkage ratio (%); *A*_0_ and *A*_1_ represent the area of the sample at the beginning and end of drying, respectively.

#### 2.6.2. Color Properties

The colorimetric values (*L**, *a**, and *b**) of the images were obtained using MATLAB software according to the following procedure: import the image, intercept the region of interest; convert the RGB color space into the CIELab color space, and calculate the color parameters [29]. The color difference (Δ*E*) between the dried samples and the control (freeze-dried samples) was calculated using Equation (10),
(10)$$\Delta E=\sqrt{{\left(L^{*}-L_{o}{}^{*}\right)}^{2}+{\left(a^{*}-a_{o}{}^{*}\right)}^{2}+{\left(b^{*}-b_{o}{}^{*}\right)}^{2}}$$
where *L*_0_*, *a*_0_*, *b*_0_* and *L**, *a**, *b** represent the color parameters of the control and dried samples, respectively.

### 2.7. GC-MS Analysis

The volatile compounds of dried chrysanthemum powder were extracted using headspace solid phase micro-extraction (HS-SPME). A total of 1 g dried chrysanthemum powder was weighed and blended with 10 μL internal standard (cyclohexanone, 95.3 μg/L), then transferred to a headspace vial (10 mL), incubated at 50 °C for 10 min and sealed. Then, an SPME fiber (75 μm CAR/PDMS) was exposed to the headspace for 40 min and inserted into the injection port of GC for desorption in the spitless mode (250 °C, 5 min).

Subsequently, the volatile compounds in the dried samples were analyzed using a GCMS-QP2020 system (Shimadzu Company, Kyoto, Japan). DB-Waxetr capillary column (30.0 m × 0.25 μm × 0.25 mm, Agilent, Santa Clara, CA, USA) was utilized to separate the samples volatiles. The ramped temperature procedure was set as follows: 40 °C for 2 min; increased 70 °C at 6 °C/min; increased to 120 °C at 4 °C/min, maintained for 3 min; increased to 230 °C at 6 °C/min, maintained for 5 min. MS conditions: injector and interface temperature, 250 °C; ion source temperature, 230 °C; ionization energy, 70 eV; mass range, 40 to 500 amu. Qualitative and semi-quantitative analysis of volatile compounds were performed using the method [30].

#### 2.7.1. Principal Component Analysis (PCA) of Volatile Compounds

PCA is a well-established data dimensionality reduction algorithm that transforms a set of potentially correlated variables into a set of uncorrelated variables through an orthogonal transformation, and the new variables after the transformation are called principal components (PCs) [31]. Essentially, PCs are linear combinations of the original variables (i.e., the concentration of volatile compounds). PCA was performed using Origin 2021 software (Origin Lab, Northampton, MA, USA) to calculate the PCs of volatile compounds in dried chrysanthemum cakes, which mainly consisted of constructing matrices, centering the measurement types and calculating PCs using singular value decomposition (SVD).

#### 2.7.2. Hierarchical Cluster Analysis (HCA) of Volatile Compounds

HCA is a cluster algorithm that analyzes data at different levels based on appropriate metric of variable distance, ultimately forming a tree-like clustering structure [32]. Specifically, based on the distances between variables, the variables with the smallest distances are merged into one cluster in an aggregation-by-aggregation manner, until finally all variables are clustered into one cluster. In this study, HCA was conducted to analyze the GC-MS data using *R* Studio version 4.1.2 (Boston, MA, USA) [33], and HCA dendrograms were generated to further visualize the relationships between the variables.

### 2.8. Determination of Phenolic Compounds

#### 2.8.1. Extraction of Phenolic Compounds

The extraction of phenolic compounds was conducted according to the method described by Major et al. [34] with slight modifications. 100 mg dried sample powder was mixed with 5 mL of 60% methanol (*v*/*v*), followed by treatment at 70 °C for 30 min in an ultrasonic water bath (40 kHz and 200 W). The suspension was centrifuged at 10,000× *g* for 15 min, and the supernatant obtained was filtered through a 0.22-μm nylon membrane for UPLC-MS analysis.

#### 2.8.2. UPLC-MS/MS Analysis of Phenolic Compounds

Chromatographic separation was carried out using an Acquity UPLC HSS T3 (2.1 × 100 mm, 1.8 μm, Waters, Milford, MA, USA) column at 30 °C. The flow rate and injection volume were set to 0.3 mL/min and 5 μL, respectively. The mobile phase comprised acetonitrile (phase A) and 0.05% formic acid water (phase B). The solvent gradient was as follows: 0–0.5 min, 10% A–90% B, 0.5–5 min, 10% A–90% B, 5–9.5 min, 30% A–70% B, and 9.5–11 min, 90% A–10% B. MS was performed using electrospray ionization source in multiple reaction mode (MRN) with the following conditions: capillary voltage +3100 V (positive mode) and −2900 V (negative mode), source temperature 120 °C, desolvation temperature 400 °C, cone gas (nitrogen) flow 50 L/h, desolvation gas (nitrogen) flow 600 L/h, collision gas (argon) flow 0.07 mL/min. The acquired data were further analyzed quantitatively using a standard calibration curve.

#### 2.8.3. Total Phenolic Content (TPC)

TPC was measured using the Folin–Ciocalteu method [35] and expressed as gallic acid equivalents (GAE). Briefly, the supernatant (0.1 mL) was blended with 1 mL of Folin-Ciocalteu reagent and 2 mL of sodium carbonate solution (15%, *v*/*v*). After dilution to 10 mL, the reaction solution was incubated in a constant temperature bath at 50 °C for 30 min in the dark. Absorbance was determined at λ = 760 nm using a spectrophotometer (TU-1810, Beijing Puekinje General Instrument Co., Ltd., Beijing, China).

#### 2.8.4. Total Flavonoid Content (TFC)

For the TFC assay [36], 0.5 mL of extract was transferred into a 10 mL volumetric flask and blended with 0.3 mL of sodium nitrite solution (5%, *v*/*v*), 0.3 mL of aluminum nitrate solution (10%, *v*/*v*) and 2 mL of sodium hydroxide (4%, *v*/*v*). The reaction solution was then diluted to 10 mL and incubated in the dark at room temperature for 15 min. The absorbance was determined at 510 nm using a spectrophotometer. The results were expressed as rutin equivalents (RE).

### 2.9. Water Binding Capacity (WBC) and Water Holding Capacity (WHC)

*WBC* and *WHC* were measured using the method as described by Su et al. [37].

*WBC*: 0.5 g of the chrysanthemum powder was mixed with 50 mL of distilled water and then the mixture was centrifuged at 8000 rpm for 5 min. The supernatant was slowly poured out, and the *WBC* was determined using Equation (11),
(11)$$WBC=\frac{M_{w}-M_{d}}{M_{d}}$$
where, *M_d_* indicates the weight of the dried chrysanthemum powder (g), *M_w_* indicates the weight of the dried chrysanthemum powder with absorbed water (g).

*WHC*: After the mixture was left for 24 h, it was centrifuged at 5000 rpm for 10 min. Similarly, *WHC* was calculated as follows:(12)$$WHC=\frac{M_{w}-M_{d}}{M_{d}}$$

### 2.10. Microstructure Analysis

The petals of chrysanthemum cakes dried using different drying methods were fixed on aluminum stubs with conductive glue. After the surfaces of the samples were sprayed with gold plating, they were observed at 500× and 1000× magnification using a scanning electron microscope (SU3500, Hitachi, Japan) with an acceleration voltage of 15 kV.

### 2.11. Statistical Analysis

The experimental results under different conditions are expressed as the mean ± standard deviation of triplicate data. Significance analysis of the differences between means were performed by Duncan’s test (*p* < 0.05) using SPSS 20.0 (SPSS Inc., Chicago, IL, USA). The experimental data were plotted using Origin 2021 software (Origin Lab, Northampton, MA, USA). Pearson correlation analysis was performed on the data with the help of Origin 2021 software. A plotted correlation diagram was used to visualize the correlation between each pair of variables. The red and blue colors represent the positive and negative interdependence (direction) between the variables, respectively. The shades of color and the size of the solid circle represent the magnitude of the absolute value of the correlation coefficient.

## 3. Results

### 3.1. Drying Kinetics

#### 3.1.1. Moisture Ratio (*MR*) and Drying Rate (*DR*)

The drying characteristic curves of chrysanthemum cakes underwent different conditions, showing that the moisture content decreased from 74.86% (wet basis) to less than 10% (wet basis) with increasing drying time (Figure 3A,C,E), with drying times varying from 1080–1560, 300–750, and 660–2040 min for HAD, IR-HAD and IR-HAD + HAD, respectively. Successive increases in temperature significantly (*p* < 0.05) reduced drying time, especially when the temperature was increased from 55 to 75 °C, IR-HAD + HAD led to a 70.54% reduction in drying time, which may be due to the enhanced thermal driving force within the chrysanthemum cake caused by the increased temperature [38]. Figure 3B,D,F indicated that the drying rate gradually reduced with a decrease in the dry basis moisture content. The constant rate period was absent during the drying process of the chrysanthemum cakes. As shown in Figure 3B, a transient rising phase in the drying rate curve was observed, which was because samples were still in the warming stage. Considering the effect of drying strategies on the drying characteristics of chrysanthemum cakes, IR-HAD with the shortest drying time was found to possess a higher drying efficiency than the other drying strategies. This may be because chrysanthemum flowers are double flowers with multiple whorls, which are not conducive to heat transfer. However, infrared radiation with penetrating ability could rapidly warm up the sample by exciting the vibration and friction of water molecules [39]. Unfortunately, the sequential drying method (IR-HAD + HAD) was not effective in improving the drying efficiency, except at 75 °C. This phenomenon may be related to the formation of a porous structure due to the rapid moisture removal from the core to the surface in the first stage (IR-HAD) [40]. However, when the samples were transferred to a relatively mild drying environment (HAD), the collapse of porous structure resulting from continuous shrinkage hindered moisture migration, thus prolonging drying time [41].

#### 3.1.2. Mathematical Modeling

As illustrated in Table 1, all four mathematical models chosen can describe the drying behavior of chrysanthemum cake during different drying processes well. Based on higher *R*^2^ values, lower *RMSE*, *SSE*, *AIC* and *BIC* values, a better fit of the Logarithmic model (0.9952 < *R*^2^ < 0.9997, 0.0065 < *RMSE* < 0.02, 8.6 × 10^−4^ < *SSE* < 3.6 × 10^−3^, −201.68 < *AIC* < −91.01, −213.48 < *BIC* < −101.28) was observed for describing the drying behavior of chrysanthemum cakes during IR-HAD and HAD. Likewise, the experimental data of IR-HAD + HAD were better fitted by the Page model (0.9908 < *R*^2^ < 0.9957, 0.0161 < *RMSE* < 0.0230, 3.6 × 10^−3^ < *SSE* < 1.2 × 10^−2^, −157.97 < *AIC* < −97.83, −163.23 < *BIC* < −104.41). Notably, the drying rate constant (*k*) exhibited an increasing trend with increasing drying temperature. However, the trends of the *a*, *b*, *k*_1_ and *n* values with drying temperature were not clear. A similar observation was reported for the drying of yam slices [42]. Regression analysis was performed to assess the correlation between the predicted and experimental data. Figure 4 showed that the predicted and experimental moisture ratio values banded along a straight line, demonstrating the applicability of the Logarithmic and Page models in the description of the dehydration process. In summary, these two models can provide a theoretical basis for optimizing the drying process of chrysanthemum cake.

#### 3.1.3. Moisture Diffusivity (*D_eff_*) and Activation Energy (*E_a_*)

As illustrated in Table 2, the calculated *D_eff_* values increased with increasing drying temperature for all three drying strategies, suggesting that high temperatures were more favorable to improve the diffusivity of water molecules. The *D_eff_* values of chrysanthemum cakes ranged from 0.48 × 10^−7^–3.79 × 10^−7^, in line with the broad range of most food materials, varying from 10^−11^ to 10^−6^ m^2^/s [43]. IR-HAD led to the highest *D_eff_* values at the same temperature, followed by HAD and IR-HAD + HAD. Furthermore, Pearson correlation analysis showed that there was a negative correlation between *D_eff_* and drying time (*r* = −0.98), which implied that the enhancement of moisture diffusivity during drying contributed to a shorter drying time.

The activation energy (*E_a_*) determines the energy required for removing 1 mol of moisture, which can be used to evaluate the energy consumption of the drying process. As shown in Table 2, the calculated *E_a_* values for chrysanthemum cakes ranged from 26.21 to 60.76 kJ/mol, which lied within the activation energy range of most foods, between 12.7 and 110 kJ/mol [44]. The highest *E_a_* value was obtained by IR-HAD, suggesting drying temperature during IR-HAD had a more significant influence on *D_eff_* values of chrysanthemum cakes than other drying strategies. The *E_a_* values obtained in the study were higher than the results reported by Wang et al. [45]. This may be attributed to the difference in variety, tissue structure, and shape of materials [46].

### 3.2. Specific Energy Consumption (SEC)

The average *SEC* values for different drying strategies of dried chrysanthemum cakes at different temperatures are shown in Figure 5. In terms of the effect of drying method on the *SEC* values, IR-HAD resulted in the lowest *SEC* values, regardless of drying temperature, which is related to the shorter drying time required for IR-HAD. Indeed, the correlation analysis showed a positive correlation between *SEC* values and drying time (*r* = 0.98), implying the reduction in drying time played a decisive role in reducing energy consumption. Notably, IR-HAD + HAD consumed much less energy than HAD at 65 and 75 °C, which may be related to the improved heating uniformity of the chrysanthemum cake surface by IR radiation and the removal of a large amount of moisture from the interior of the chrysanthemum cakes during the first drying stage (IR-HAD). As for the effect of drying temperature on *SEC* values, it can be found that *SEC* values show an overall decreasing trend with increasing drying temperature. Regardless of the drying method, the lowest *SEC* values were obtained when drying temperature was 75 °C. However, the *SEC* values of IR-HAD at 65 °C was higher than that at 55 °C. This was most likely because the energy consumed to heat the air was higher than the energy saved by reducing the drying time. Similar results were also reported by Wang et al. [47].

Overall, IR-HAD is more suitable than other drying strategies for the industrial production of dried chrysanthemum cakes in terms of dehydration efficiency and production cost.

### 3.3. Effect of Drying Strategies on Shrinkage Ratio (SR)

The calculated *SR* values of dried chrysanthemum cakes under different drying conditions are illustrated in Table 2, which varied from 16.26 to 34.10%. For chrysanthemum, the marginal florets appeared irregularly curled during drying because of its sparse parenchyma, and the undeveloped protective tissue led to the collapse of the surface structure. Meanwhile, the space previously occupied by moisture was constantly emptied and inflated, finally inducing shrinkage of the overall structure [48].

Regarding the impact of drying methods on the *SR* values, samples dried with HAD have higher *SR* values, while samples dried with IR-HAD have lower *SR* values. This suggested that the continuous shrinkage of the tissue structure could be inhibited by shortening the drying time. In addition, the shrinkage was exacerbated by increasing temperature, which may be related to the glass transition effect [49]. Interestingly, the *SR* values of IR-HAD decreased with increasing temperature. This phenomenon could be explained that the rapid removal of moisture in the initial stage of drying resulted in the formation of mechanical stabilization on the surface [50]. Similar results were also observed during IR-HAD + HAD.

### 3.4. Effect of Drying Strategies on Color Properties

The color parameters of chrysanthemum cakes dried under different conditions are shown in Table 2. *L**, *a** and *b** values of dried samples ranged from 64.91–80.63, 0.67–3.42 and 9.19–15.77, respectively. Compared to the control, the different drying strategies reduced the *L** values, and increased the *a** and *b** values. The reduction in lightness was possibly linked to the generation of dark pigments caused by the Maillard reaction [51]. Remarkably, samples dried by the sequential drying method exhibited higher *L** values than those dried by HAD and IR-HAD at the same temperature, which indicated that IR-HAD + HAD could yield brighter products.

In fresh flower, the carotenoids that cause the yellow reflection of central disc florets mainly are located in the chromoplasts. Thermal treatment destroys the cellular architecture and facilitates the entry of carotenoids into the intercellular space [52], thus resulting in more yellow petals. Indeed, the *a** and *b** values exhibited an increasing trend with increasing temperature in all cases, suggesting that high temperature exacerbated the color deterioration of samples. This may be because chrysanthemums contain amino compounds (proteins, amino acids, etc.) and hydroxyl compounds (reducing sugars, etc.), which undergo Maillard reactions under high temperature conditions, resulting in the formation of a reddish-brown substance [53]. Similarly, Senadeera et al. [54] reported that high temperatures and prolonged durations were the dominant factors causing color deterioration.

For drying temperatures of 55–75 °C, color difference (Δ*E*) values of dried chrysanthemum cakes ranged from 5.02 ± 0.04 to 19.90 ± 1.07. Comparatively, the sequential drying method (IR-HAD + HAD) presented lower Δ*E* values than HAD and IR-HAD for chrysanthemum cakes, suggesting that the color attributes of chrysanthemum cakes were well preserved by IR-HAD + HAD. This result can be explained by the fact that the sequential drying method facilitated the effective redistribution of heat within the material, ultimately leading to products with superior color attributes.

### 3.5. Effect of Drying Strategies on Volatile Compounds

As shown in Table 3, A total of 48 volatile compounds, including terpenes (25), ketones (5), esters (5), hydrocarbons (4), aldehydes (4), alcohols (3), furans (1), and phenols (1), were identified in four detected samples. Thereinto, terpenes, ketones, and esters were the main volatile compounds. Compared to the freeze-dried samples (38), the number of volatile compounds (Figure 6A) showed a slight increase in HAD (44), IR-HAD + HAD (41). However, the concentration of most volatile compounds in the samples obtained using different drying strategies, especially for samples dried using HAD (Figure 6B).

Terpenes accounted for more than 50% of the total aroma quantity in detected chrysanthemum samples. Therefore, terpenes can be considered as the major contributors to the flavor of chrysanthemums. These terpenes emitted floral and fruit odors due to their sensory stimulation effects. For instance, D-limonene was described as having a lemon-like smell, and β-caryophyllene and linalool emitted woody and floral odors. In addition, the results of the current study indicated that the three drying methods profoundly reduced the total content of terpenes, compared to the control. This may be due to the degradation of terpenes caused by the heat and energy generated during drying. Individual terpenoids differed in molecular size and structure leading to differences in their volatility and thermal degradation properties which can explain the differences in the retention levels of the different drying treatments [55]. As shown in Figure 6B, IR-HAD exhibited a higher preservation level of terpenes than HAD and IR-HAD + HAD, which may be related to its shorter drying time, implying that shortening the time the sample was exposed to heating could effectively reduce the volatile loss. β-sesquiphellandrene, α-zingiberene, γ-muurolene and α-curcumene were observed to be the predominant aromatic constituents isolated from chrysanthemum resulting from their extremely high concentrations. Their pharmaceutical properties, such as anti-inflammatory, anti-cancer and antimicrobial, were of great benefit to human health and have attracted increasing attention from researchers in the medical community [56]. Thus, most of terpenes contained in chrysanthemum are essential components of traditional Chinese medicine, which directly determines the commercial value of the products.

C10 ketones were the dominant ketone components in chrysanthemum samples, and all ketone components were detected and preserved better in the FD samples (Table 3). Compared with FD, the content of ketone compound in samples dried with different drying strategies decreased drastically, especially for HAD alone. This phenomenon may be because thermal process destroyed C10 ketones and strongly affect the reactions associated with them [57]. Samples dried using IR-HAD possessed higher retention levels of ketones, suggesting that drying time may be a critical factor affecting the C10 ketone content.

As products of the esterification reaction, esters constitute an essential component of the volatile compounds of chrysanthemum samples. Five ester compounds (trans-chrysanthenyl acetate, bornyl acetate, methyl caprate, benzyl pentanoate and benzyl isovalerate) were detected in the dried samples, they emitted floral, fruity and sweet aromas. There was a noticeable difference in the retention level of esters, and IR-HAD yielded higher ester compound content than the other two strategies (Figure 6B). Similarly, all aldehydes detected exhibited higher accumulation levels in IR-HAD samples than in the other two drying strategies. Benzaldehyde and 3-hydroxybenzaldehyde were the most abundant aldehyde compounds in the samples. Furfural and 4-iPr-benzaldehyde were absent in the freeze-dried samples, presumably they may have been formed during thermal processing. These compounds may stem from chemical reactions related to α-dicarbonyl compounds generated by the Maillard reaction [58]. Similarly, furan was also a product of the Maillard reaction. Dihydro-β-agarofuran was the only furan compound detected in chrysanthemum samples, and its concentration in IR-HAD + HAD samples was higher than the other two drying strategies. These results suggested that the Maillard reaction contributed to enriching the aroma category of dried chrysanthemum cakes during drying.

Regarding the other volatile compounds, 3-methyl-3-cyclohexen-1-ol and 2,4-dimethyl-4-octanol were alcohols, were detected only after thermal treatment, and shorter drying times or milder drying conditions can inhibit volatilization, degradation and conversion of alcohols. When compared to hydrocarbons, 1,3,5,7-cyclooctatetraene was found to be the most dominant hydrocarbon compound (Table 3). The total number of hydrocarbons in the HAD samples was higher, whereas IR-HAD samples had a higher retention level of hydrocarbons. For phenols, butylated hydroxytoluene was only detected in the HAD and IR-HAD samples, and it was a commonly used food additive.

The aroma profile of chrysanthemum cakes is in a complex and dynamic state during drying. The evaporation and degradation of heat-sensitive volatile compounds in samples caused by heating are not the only determinants of their content changes underwent different drying conditions, and the dynamic chemical transformation among different volatile compounds in samples is also crucial. Overall, although different drying strategies promoted the generation of some volatile compounds, the total content of volatile compounds in dried chrysanthemum cakes showed a considerable reduction compared to freeze-dried samples. This phenomenon may be because the decomposition of the heat-sensitive components caused by high temperature generated during drying. These findings were confirmed by Pearson correlation analysis, the retention levels of volatile compounds, except terpenes, were found to be significantly (*p* < 0.05) negatively correlated (−0.90 < *r* < −0.98) with drying time. In terms of the effect of drying strategies on the volatile compounds, IR-HAD-dried samples were found to have the highest concentration of total volatile compounds, while HAD-dried samples exhibited a diverse aroma profile. To make full use of the value of medicinal chrysanthemums, different drying strategies can be employed when targeting different industrial needs: HAD is suitable for enriching aroma profile; IR-HAD is beneficial for preserving aroma components.

### 3.6. HCA and PCA of Volatile Compounds

PCA was carried out on the GC-MS dataset to distinguish dried chrysanthemum cakes from different drying methods (Figure 7). Two principal components, PC1 (72.1%) and PC2 (17.4%), which explained 89.5% of the total variance. Drying methods in different quadrants on the PCA biplot indicated significant differences, and all quantified volatile compounds were further distinguished into clusters corresponding to the drying methods and similarities. PC1 explained the variance associated with the differences between freeze-dried and heat-treated samples (HAD, IR-HAD and IR-HAD+HAD). In the same manner, PC2 explained the data variance between different drying strategies, and discriminated dried samples subjected to HAD, IR-HAD and IR-HAD + HAD in the second and third quadrants of the PAC biplot. The particular proximity between IR-HAD and volatile compounds nos. 4, 9, 10, and 43 signified the power of ylangene, γ-muurolene, α-zingiberene and furfural to discriminate IR-HAD from other dried samples. Additionally, HAD located in the third quadrant presented the highest scores on these principal components (PC1, PC2), which may be attributed to its low aroma concentration. Likewise, volatile compounds nos. 1–, 5–8, 13, 16–18, 20, 22, 25–35, 37, 40, 46, 48 were correlated with the freeze-dried samples. Through PCA, we were able to purposefully find the potential volatile compounds of dried chrysanthemum cakes, and thus gain more insight into the differential information present in these samples, which would help us to have a more comprehensive evaluation of the aroma profile of chrysanthemum cakes dried by different methods.

To further validate the findings obtained from PCA, an HCA dendrogram was plotted to reveal the intrinsic relationship between samples dried by different methods, using a systematic clustering model with Euclidean distance as the measurement criterion. Furthermore, a heat map was generated to visualize the distribution of volatile compounds in dried samples. As shown in Figure 6C, the four dried chrysanthemum cakes were evidently classified into three clusters at a Euclidean distance of 5.86. HAD and IR-HAD + HAD belonged to one group, while both FD and IR-HAD belonged to different groups. These results were consistent with those observed from PCA. In addition, the heat map showed that concentration of volatile compounds in FD-dried samples exhibited a darker red color, implying that FD provided higher concentrations of volatile compounds compared to other drying methods. Interestingly, HAD-dried and IR-HAD + HAD-dried samples were not differentiated, implying that the aroma profile of the samples dried by both drying methods were relatively similar. However, IR-HAD-dried samples were separated from the HAD-dried and IR-HAD + HAD-dried samples, and the concentration of volatile compounds in IR-HAD-dried samples showed a darker red and a lighter blue color, indicating a higher level of volatile compounds in IR-HAD-dried samples. These findings suggested that IR-HAD possessed a considerable potential for preserving the volatile compounds in dried chrysanthemum cakes.

### 3.7. Effect of Drying Strategies on Physicochemical Properties

#### 3.7.1. Water Binding Capacity (*WBC*) and Water Holding Capacity (*WHC*)

*WBC* and *WHC* represent the water content that can be bound and retained by the material under certain conditions, respectively [59]. Thus, they can be considered as indices for assessing the degree of damage to dried products. The impacts of drying conditions on the *WHC* and *WBC* of dried chrysanthemum cakes are shown in Table 2. Compared with FD, the other three drying methods resulted in lower *WBC* and *WHC* values, which may be derived from the collapse of the supporting structure of chrysanthemum cakes caused by the thermal process. IR-HAD presented higher *WBC* and *WHC* values due to a higher moisture migration gradient compared to HAD and IR-HAD+HAD, implying that the drying process with high intensity caused tissue collapse and loss of cell integrity during IR-HAD. Notably, Pearson correlation analysis (Figure 8) also showed that *WBC* had a negative correlation with shrinkage (*r* = −0.94). Furthermore, all samples dried at 65 °C had higher *WHC* and *WBC* values than other temperatures, regardless of drying method, indicating that the appropriate drying temperature could effectively reduce damage to products. The results showed that the drying method, temperature and their interactions had significant (*p* < 0.05) implications for *WHC* and *WBC*.

#### 3.7.2. Effect of Drying Strategies on Phenolic Compounds

The UPLC-MS analysis results of chlorogenic acid and luteolin in chrysanthemum dried by different methods are presented in Figure 9. The drying methods had significant (*p* < 0.05) differences in the retention level of phenolic compounds, and freeze-dried samples exhibited the highest contents of chlorogenic acid and luteolin, which can be explained by the fact that the thermal sensitivity of phenolic compounds was responsible for degradation. Compared to HAD alone, IR-HAD + HAD processed samples had double the chlorogenic acid content, suggesting that the introduction of infrared radiation significantly enhanced the accumulation of phenolic compounds. Similarly, Zhou et al. [60] reported that infrared drying had a lower degradation rate of phenolic compounds than hot air drying. Additionally, Juániz et al. [61] found the wall and cell rupture due to thermal treatment increased the bioavailability of phenolic components. Interestingly, Pearson correlation analysis (Figure 8) showed a positive correlation between chlorogenic acid content and lightness (*L**) values (*r* = 0.67), this phenomenon could be explained by the fact that IR-HAD inhibited the production of brown pigments caused by the degradation or conversion of chlorogenic acid isomers, thus increasing *L** values [62].

TPC and TFC of dried chrysanthemum cakes under different drying conditions are presented in Table 2, ranging from 26.91 to 52.31 GAE/g DW and from 86.26 to 133.49 RE/g DW, respectively. Compared to FD, appropriate conditions (temperature or time) were conducive to increasing the bioactive composition content. This may be because thermal treatment disrupted the plant cell wall and facilitated the release of bound phenolics in the cell wall matrix, thus increasing the concentration of phenolics [63]. Regarding the influence of drying strategies on TPC, HAD led to the lowest TPC value (26.91 ± 0.40 mg GAE/g DW), indicating that prolonged exposure to thermal energy was detrimental to the accumulation of TPC. As expected, the highest TPC value (52.31 ± 2.49 mg GAE/g DW) was obtained by IR-HAD. Similarly, Wen et al. [64] found that high molecular phenolics could be transformed into low molecular phenolics due to the covalent bond breakage of polymeric polyphenols during infrared drying, which increased the retention of phenolic compounds. Surprisingly, IR-HAD + HAD yielded higher retention of TPC than HAD alone, benefiting from the introduction of infrared radiation.

Thermal processing may disrupt vacuoles which are the primary sites for flavonoid compounds [65], facilitating the extraction of flavonoid compounds from products. However, prolonged exposure of the samples to heat resulted in oxidation reactions and degradation of flavonoid compounds. Considering HAD, the successive temperature increase caused a corresponding increase in TFC, which implied that the oxidation and degradation losses of TPC were reduced owing to the shortening of the drying time. However, for IR-HAD and IR-HAD + HAD, excessive drying temperature (75 °C) led to a significant (*p* < 0.05) reduction in TFC. Similarly, Xu et al. [66] demonstrated that high temperatures and long drying times were detrimental to the retention of TFC in cabbage. Interestingly, the introduction of infrared radiation had no positive effect on the retention of TFC, and the lowest value (86.26 ± 0.49 mg RE/g DW) of TFC was obtained at 55 °C using IR-HAD + HAD. These results suggested that the loss of phenolic compounds in chrysanthemum cakes may be due to the synergistic impact of the drying temperature and time [67]. Notably, the retention level of TFC was positively correlated (*r* = 0.85) with *L** values, suggesting that the color deterioration may be related to the degradation of phenolic compounds [68].

### 3.8. Effect of Drying Strategies on Microstructure

The petals of dried chrysanthemum cakes were used for SEM observations. As illustrated in Figure 10, it could be observed that the petal epidermal cells obtained by different drying methods showed different degrees of shrinkage. For FD, the cells of the samples had regular spatial shapes, similar sizes, and slight collapse. Additionally, a relatively low degree of shrinkage was observed, indicating that the morphology of the cells was well preserved by FD. Following HAD treatment, the cells exhibited a compact state due to squeezing of the cell walls. More severe cellular shrinkage and collapse occurred compared with the freeze-dried samples. This phenomenon suggested that low drying efficiency may result in prolonged tissue contraction time. The cellular structure of the samples dried by IR-HAD shown an obvious collapse and shrinkage. This phenomenon could be attributed to the vapor pressure difference between surroundings and interior of the samples during drying. The higher drying rate of IR-HAD caused internal cell rupture, which contributed to the extraction of phenolic compounds from the chrysanthemum. Notably, the distribution inhomogeneity of cells was improved compared with that of HAD. Similarly, samples dried by IR-HAD + HAD presented a more uniform cell distribution than those dried by HAD alone, which may be related to the tissue expansion induced by IR-HAD. This was conducive to reducing the degree of cell shrinkage.

## 4. Conclusions

In this study, we comparatively investigated the effects of different drying strategies (HAD, IR-HAD, IR-HAD + HAD) on the drying kinetics, shrinkage, color, aroma profiles, phenolic compounds, water binding capacity, water holding capacity and microstructure of chrysanthemum cake. Results showed that drying method and temperature significantly influenced the drying behavior of chrysanthemum cakes, mainly in terms of drying time, drying rate, moisture diffusion and activation energy. The Logarithmic and Page models exhibited a better fit in describing the dehydration process of chrysanthemum cake based on higher *R*^2^, lower *RMSE*, *SSE*, *AIC* and *BIC* values. IR-HAD had excellent potential to reduce energy consumption, and improve shrinkage degree, *WBC*, *WHC* and accumulation of phenolic compounds of dried chrysanthemum cakes. This may be related to the rupture of the cellular structure caused by infrared energy. Interestingly, the sequential drying method (IR-HAD + HAD) employed in the drying of chrysanthemum cake showed better color protection effects. Furthermore, GC-MS analysis suggested that different drying strategies were conducive to enriching the aroma components of samples compared to freeze drying, and terpenes were the most dominant volatile compounds in dried samples. PCA and HCA could effectively distinguish the aroma profile of chrysanthemum cakes dried by different methods based on the similarities and differences of volatile compounds. The heat map showed that the concentration of volatile compounds in IR-HAD-dried samples exhibited a darker red color and lighter blue color than that of samples dried by other drying strategies, suggesting that IR-HAD has better aroma retention ability. Overall, considering the drying efficiency, energy consumption, retention levels of phenolic and volatile compounds, and microstructure, combined infrared and hot air drying (IR-HAD) is the most recommended method for industrial production of dried medicinal chrysanthemum products.

## Figures and Tables

**Figure 1 foods-11-02240-f001:**
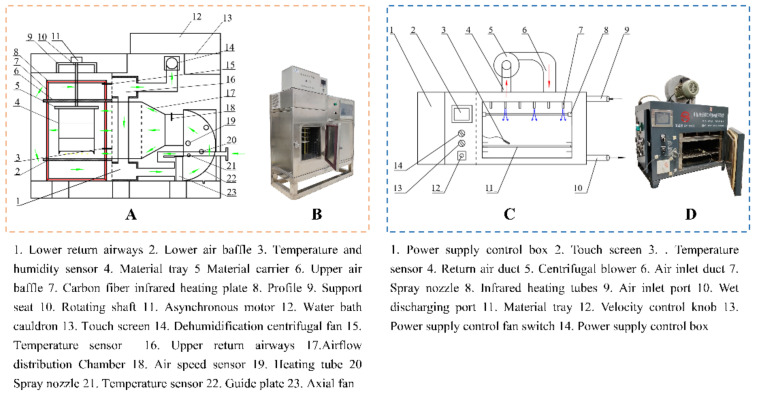
The schematic and physical diagram of dryers (The arrows represent the direction of the airflow).

**Figure 2 foods-11-02240-f002:**
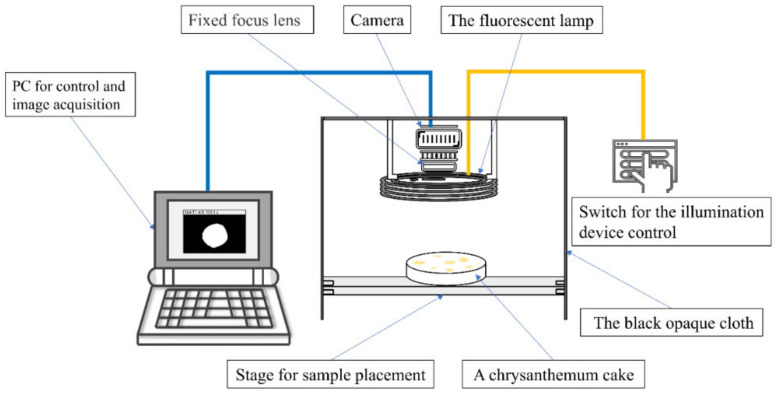
The schematic of the custom-built computer vision system.

**Figure 3 foods-11-02240-f003:**
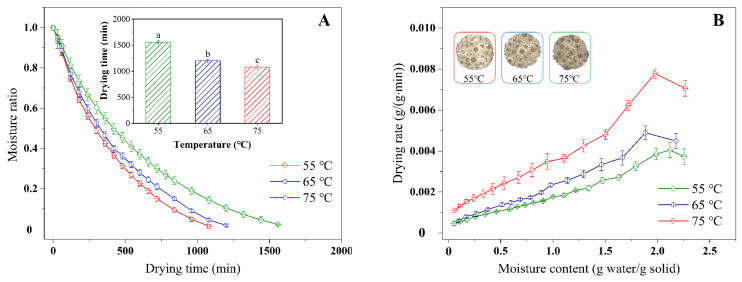
Moisture ratio (**A**,**C**,**E**) and drying rate (**B**,**D**,**F**) of chrysanthemum cakes under different drying methods: (**A**,**B**): hot air drying (HAD); (**C**,**D**): combined infrared and hot air drying (IR-HAD); (**E**,**F**): sequential IR-HAD and HAD (IR-HAD + HAD), different lowercase letters reveal significant differences (*p* < 0.05).

**Figure 4 foods-11-02240-f004:**
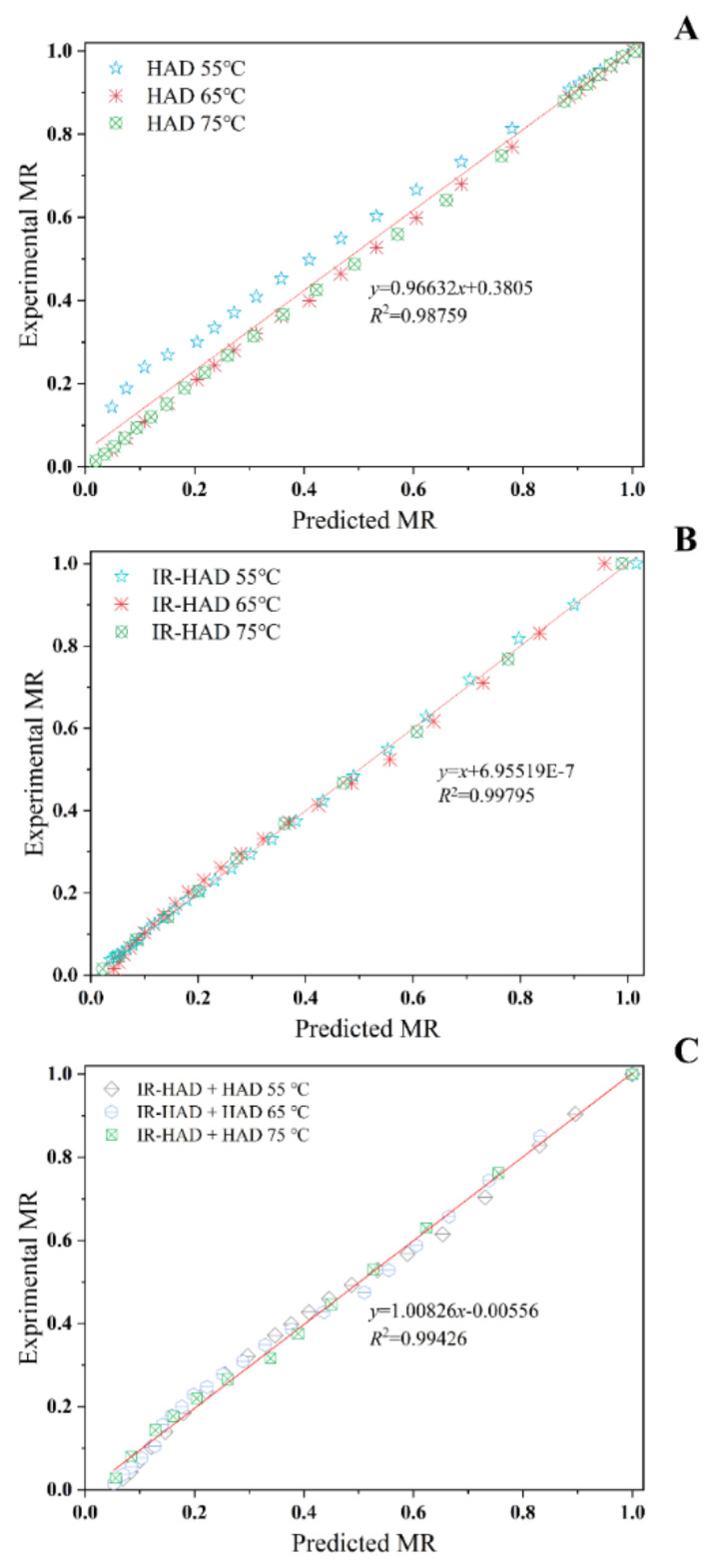
Predicted moisture ratio by Logarithmic model (**A**) versus experimental moisture ratio under hot air drying (HAD); predicted moisture ratio by Logarithmic model (**B**) versus experimental moisture ratio under combined infrared and hot air drying (IR-HAD); and predicted moisture ratio by Page model (**C**) versus experimental moisture ratio under sequential IR-HAD and HAD.

**Figure 5 foods-11-02240-f005:**
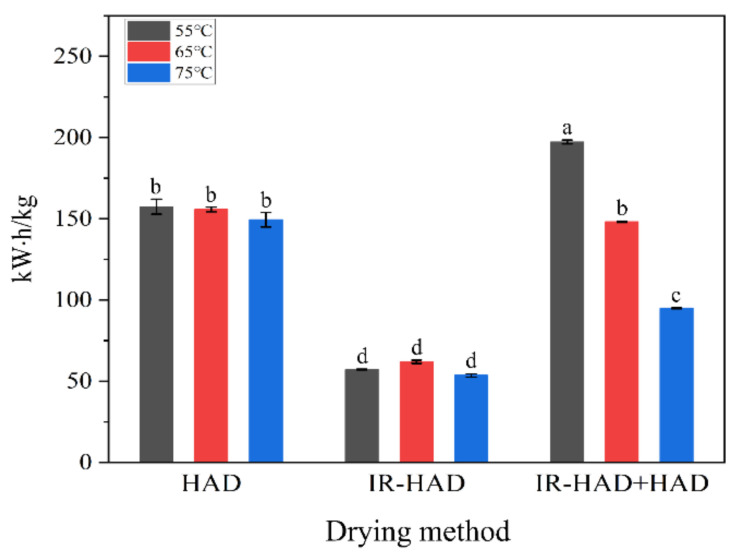
Specific energy consumption for dried chrysanthemum cakes from different drying strategies. Different lowercase letters on the bar graph represent significant differences (*p* < 0.05).

**Figure 6 foods-11-02240-f006:**
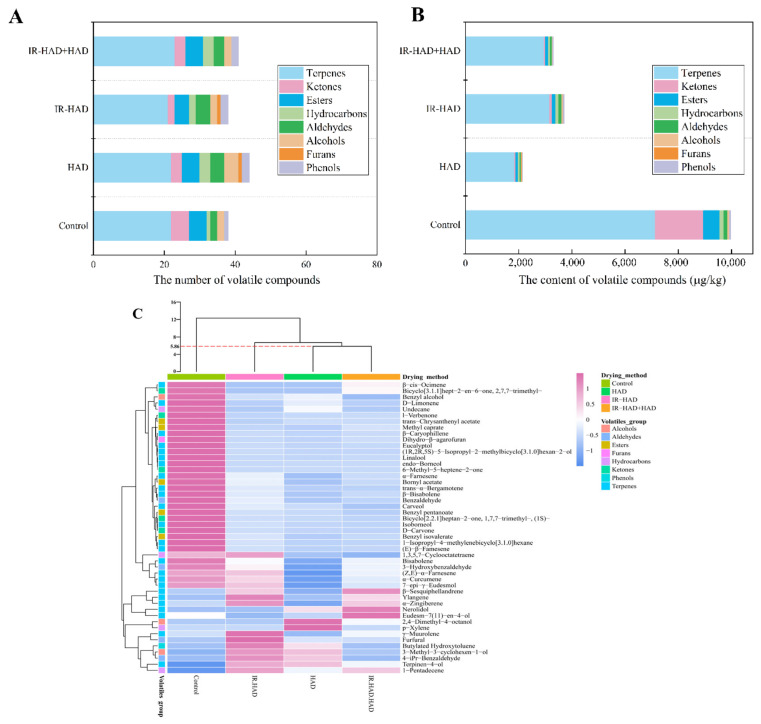
The total number (**A**), total content (**B**), hierarchical clustering analysis (HCA) and heat map (**C**) of 48 volatiles detected in chrysanthemum cakes dried by different methods: Control: freeze-drying; HAD, hot air drying at 65 °C, IR-HAD, combined infrared and hot air drying at 65 °C, IR-HAD + HAD, sequential IR-HAD and HAD at 65 °C.

**Figure 7 foods-11-02240-f007:**
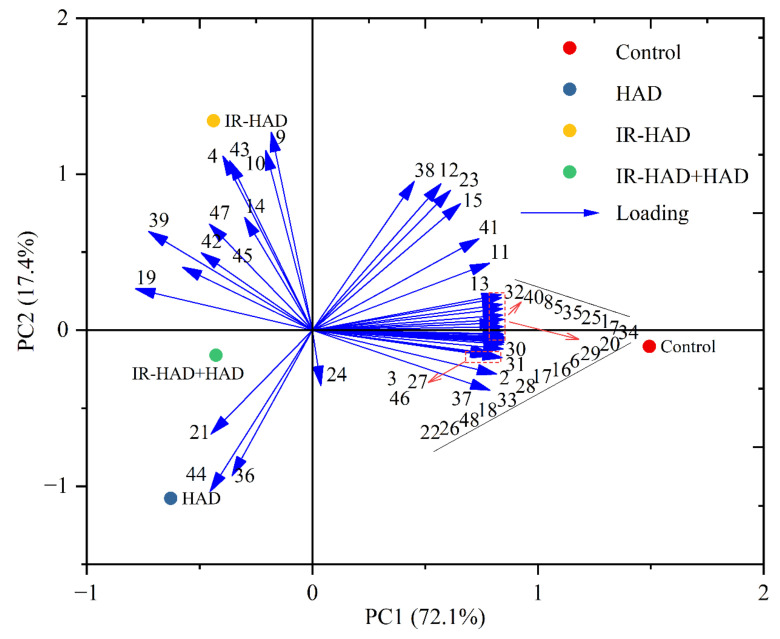
PCA biplot of volatile compounds in chrysanthemum cakes dried by different methods: Control: freeze-drying; HAD, hot air drying at 65 °C, IR-HAD, combined infrared and hot air drying at 65 °C, IR-HAD + HAD, sequential IR-HAD and HAD at 65 °C (aroma compound numbers correspond to the numbers in Table 3).

**Figure 8 foods-11-02240-f008:**
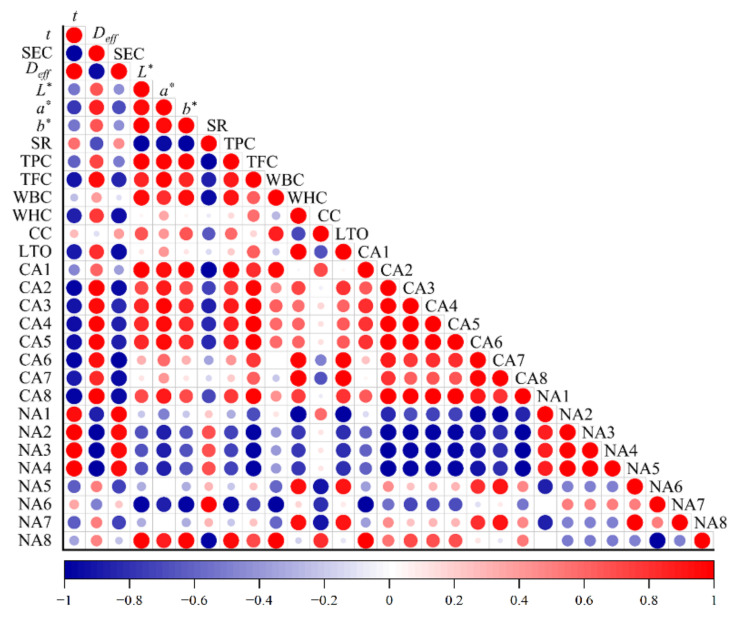
Pearson correlation analysis on various parameters of chrysanthemum cakes dried by different methods. *t*: drying time; *D_eff_*: effective moisture diffusivity; *SEC*: specific energy consumption; *L**: lightness; *a**: redness/greenness; *b**: yellowness/blueness; *SR*: shrinkage ratio; TPC: total phenolic compounds: TFC: total flavonoid content; *WBC*: water binding capacity; *WHC*: water holding capacity; CC: chlorogenic acid; LTO: luteolin; CA: the content of volatile compounds; NA: the number of volatile compounds; 1: terpenes; 2: ketones; 3: esters: 4: hydrocarbons; 5: aldehydes; 6: alcohols; 7: furans; 8: phenols.

**Figure 9 foods-11-02240-f009:**
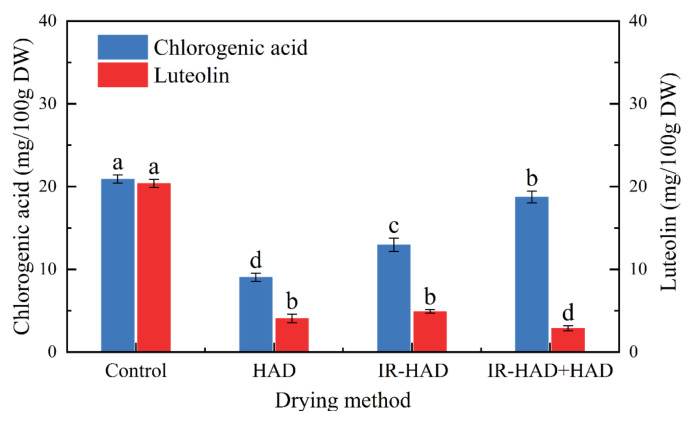
The contents of chlorogenic acid and luteolin in chrysanthemum cakes dried by different methods. Control: freeze-drying; HAD: hot air drying at 65 °C; IR-HAD: combined infrared and hot air drying at 65 °C; IR-HAD + HAD: sequential IR-HAD and HAD at 65 °C. Different lowercase letters (a–d) represent significant differences between the values of different drying methods (*p* < 0.05).

**Figure 10 foods-11-02240-f010:**
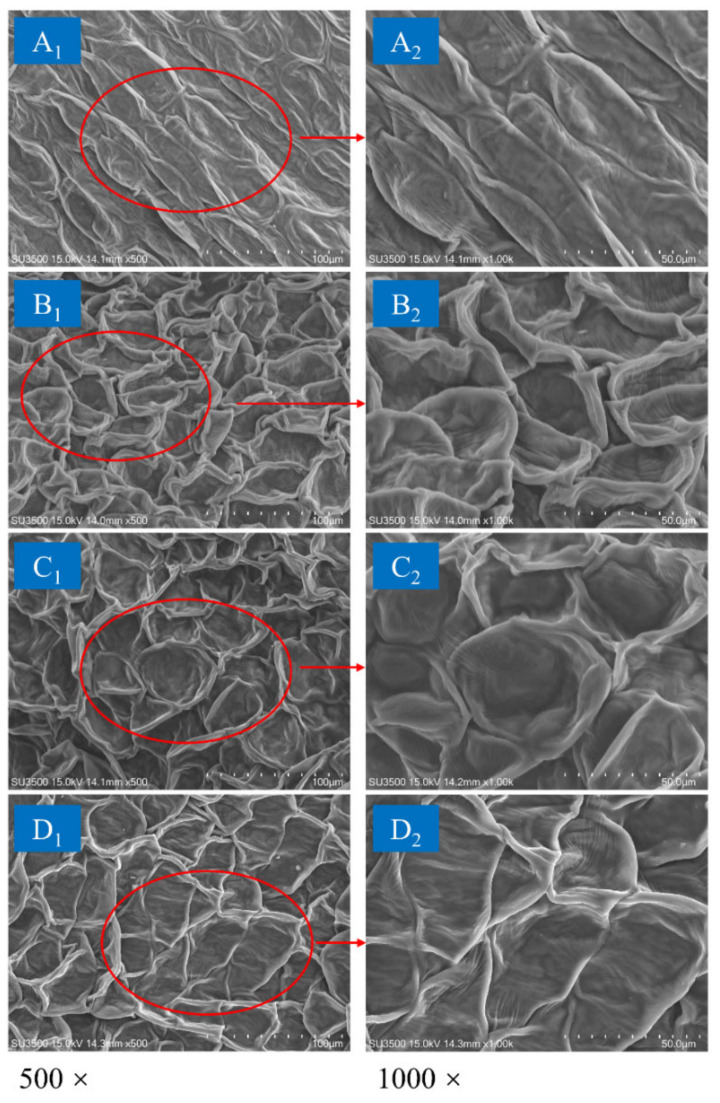
Scanning electron microscope images of dried chrysanthemum at 65 °C by different drying methods: (**A**): freezing drying; (**B**) hot air drying (HAD); (**C**) combined infrared and hot air drying (IR-HAD); (**D**): sequential IR-HAD and HAD (IR-HAD + HAD).

**Table 1 foods-11-02240-t001:** Statistical results from different mathematical models.

Model Name		HAD			IR-HAD			IR-HAD + HAD		
		55 °C	65 °C	75 °C	55 °C	65 °C	75 °C	55 °C	65 °C	75 °C
Two-Term	*a*	0.5042	0.5069	0.5180	0.9650	0.9167	1.018	0.8539	0.8291	0.8373
*MR* = *aexp* (*−k × t*) + *bexp* (*−k*_1_ × *t*)	*k*	0.0016	0.0021	0.0027	0.0052	0.0044	0.0091	0.0017	0.0024	0.0048
	*b*	0.5041	0.5068	0.5177	0.1164	0.2278	0.6357	0.8329	0.6127	0.3396
	*k_1_*	0.0016	0.0021	0.0027	0.9228	0.0710	0.7537	0.6072	0.4640	0.7922
	*R* ^2^	0.9972	0.9987	0.9960	0.9940	0.9955	0.9791	0.9804	0.9796	0.9825
	*RMSE*	0.0198	0.0121	0.0222	0.0185	0.0154	0.0356	0.0355	0.0349	0.0298
	*SSE*	6.3 × 10^−3^	2.5 × 10^−3^	9.3 × 10^−3^	5.5 × 10^−3^	3.8 × 10^−3^	7.6 × 10^−3^	2.0 × 10^−2^	2.2 × 10^−2^	7.1 × 10^−3^
	*AIC*	−118.85	−144.87	−157.16	−206.99	−176.85	−58.05	−132.09	−138.22	−79.09
	*BIC*	−127.38	−152.94	−163.98	−213.06	−183.67	−73.12	−139.16	−145.04	−90.73
Page	*k*	0.0011	0.0018	0.0010	0.0075	0.0062	0.0061	0.0110	0.0155	0.0174
*MR* = *exp* (−*k* × *t^n^*)	*n*	1.0626	1.0997	1.1493	0.9209	0.9741	1.0779	0.7310	0.7276	0.7800
	*R* ^2^	0.9982	0.9987	0.9976	0.9953	0.9959	0.9962	0.9922	0.9908	0.9957
	*RMSE*	0.0146	0.0101	0.0180	0.0191	0.0187	0.0235	0.0230	0.0240	0.0161
	*SSE*	5.1 × 10^−3^	3.4 × 10^−3^	6.4 × 10^−3^	7.3 × 10^−3^	6.7 × 10^−3^	3.9 × 10^−3^	9.5 × 10^−3^	1.2 × 10^−2^	3.6 × 10^−3^
	*AIC*	−130.05	−146.62	−171.80	−205.54	−170.79	−77.96	−154.31	−157.97	−97.83
	*BIC*	−135.82	−152.26	−177.06	−210.55	−176.05	−85.31	−159.64	−163.23	−104.41
Logarithmic	*a*	1.0752	1.0519	1.1001	0.9751	1.0048	1.0807	0.8401	0.8209	0.8373
*MR* = *aexp* (−*k* × *t*) + *b*	*k*	0.0015	0.0020	0.0021	0.0044	0.0047	0.0073	0.0018	0.0026	0.0047
	*b*	−0.0813	−0.0441	−0.1316	−0.0183	−0.0374	−0.1166	0.0228	0.0225	0.0252
	*R* ^2^	0.9997	0.9997	0.9996	0.9952	0.9977	0.9993	0.9826	0.9812	0.9846
	*RMSE*	0.0065	0.0078	0.0074	0.0200	0.0142	0.0141	0.0335	0.0335	0.0236
	*SSE*	8.6 × 10^−4^	7.8 × 10^−4^	1.1 × 10^−3^	7.6 × 10^−3^	3.6 × 10^−3^	7.4 × 10^−4^	1.9 × 10^−2^	2.1 × 10^−2^	1.2 × 10^−2^
	*AIC*	−156.83	−169.76	−207.52	−201.68	−181.44	−91.01	−136.66	−142.64	−77.84
	*BIC*	−163.77	−176.44	−213.48	−207.19	−187.40	−101.28	−142.77	−148.60	−86.45
Henderson-Pabis	*a*	1.011	1.008	1.025	0.9645	0.9332	1.018	0.8539	0.8291	0.8373
*MR* = *aexp* (−*k* × *t*)	*k*	0.0017	0.0022	0.0026	0.0052	0.0045	0.0091	0.0016	0.0024	0.0048
	*R* ^2^	0.9973	0.9984	0.9964	0.9947	0.9956	0.9843	0.9826	0.9816	0.9860
	*RMSE*	0.0174	0.0138	0.0211	0.0175	0.0152	0.0294	0.0335	0.0331	0.0266
	*SSE*	6.3 × 10^−3^	2.5 × 10^−3^	9.3 × 10^−3^	5.5 × 10^−3^	4.2 × 10^−3^	7.6 × 10^−3^	2.0 × 10^−2^	2.2 × 10^−2^	7.1 × 10^−3^
	*AIC*	−126.46	−152.16	−163.58	−212.90	−181.07	−70.62	−138.68	−144.64	−89.00
	*BIC*	−132.23	−157.79	−168.83	−217.91	−186.32	−77.97	−144.01	−149.89	−95.58

HAD, hot air drying; IR-HAD, combined infrared and hot air drying; IR-HAD + HAD, sequential IR-HAD and HAD.

**Table 2 foods-11-02240-t002:** Effective moisture diffusivity (*D_eff_*), activation energy (*E_a_*)color parameters (*L**, *a**, *b**), color difference (Δ*E*), shrinkage ratio (*SR*), water binding capacity (*WBC*), water holding capacity (*WHC*), total phenolic content (TPC), and total flavonoid content (TFC) of chrysanthemum cakes at different drying methods.

Drying Methods	Temperature	*D_eff_*	*E* _a_	Color Parameters				SR	WBC	WHC	TPC	TFC
	(°C)	(10^−7^ m^2^∙s^−1^)	kJ/mol	*L**	*a**	*b**	∆*E*	(%)	(g/g)		(mg GAE/g DW)	(mg RE/g DW)
Control	-	-		80.63 ± 0.27 ^a^	0.93 ± 0.10 ^d^	9.19 ± 0.32 ^j^	-	16.26 ± 0.56 ^g^	8.35 ± 0.68 ^a^	7.87 ± 0.13 ^a^	40.43 ± 2.66 ^cde^	119.32 ± 3.04 ^c^
HAD	55	0.61	26.21	79.08 ± 0.13 ^ab^	1.08 ± 0.13 ^d^	14.06 ± 0.25 ^cde^	5.02 ± 0.04 ^e^	26.38 ± 1.30 ^cde^	4.73 ± 0.17 ^f^	6.73 ± 0.30 ^bcd^	26.91 ± 0.40 ^g^	100.43 ± 1.58 ^e^
	65	0.82		73.96 ± 0.78 ^cd^	1.27 ± 0.29 ^d^	13.21 ± 0.68 ^de^	8.21 ± 0.64 ^cd^	26.85 ± 2.33 ^bcd^	5.11 ± 0.03 ^ef^	6.98 ± 0.65 ^abc^	44.51 ± 0.28 ^bc^	110.99 ± 0.97 ^d^
	75	1.07		64.91 ± 3.07 ^e^	1.96 ± 0.54 ^c^	13.69 ± 1.04 ^cde^	16.44 ± 2.93 ^b^	31.04 ± 2.60 ^ab^	4.74 ± 0.64 ^f^	6.02 ± 0.28 ^cde^	35.07 ± 3.79 ^ef^	118.81 ± 1.09 ^c^
IR-HAD	55	1.01	60.76	78.18 ± 0.47 ^ab^	1.59 ± 0.09 ^cd^	14.98 ± 0.17 ^bc^	6.46 ± 0.14 ^de^	26.40 ± 1.27 ^cde^	6.06 ± 0.57 ^cde^	6.45 ± 0.12 ^bcd^	45.69 ± 1.23 ^bc^	121.45 ± 1.75 ^c^
	65	1.67		76.56 ± 0.48 ^bc^	1.65 ± 0.06 ^cd^	14.59 ± 0.26 ^bc^	6.28 ± 0.30 ^de^	22.19 ± 4.26 ^ef^	6.83 ± 0.09 ^bc^	7.69 ± 0.44 ^ab^	52.31 ± 2.49 ^a^	133.49 ± 2.91 ^a^
	75	3.64		66.71 ± 2.83 ^e^	3.42 ± 1.24 ^b^	15.77 ± 0.70 ^ab^	14.73 ± 2.67 ^b^	21.62 ± 1.65 ^f^	6.50 ± 0.45 ^bc^	6.97 ± 0.35 ^abc^	34.55 ± 3.73 ^ef^	114.68 ± 0.61 ^cd^
IR-HAD + HAD	55	0.48	52.44	79.39 ± 1.11 ^ab^	0.67 ± 0.04 ^d^	11.84 ± 0.30 ^f^	5.00 ± 0.28 ^e^	28.31 ± 1.90 ^bc^	5.44 ± 0.46 ^def^	5.31 ± 0.33 ^e^	42.15 ± 0.11 ^de^	86.26 ± 0.49 ^f^
	65	0.81		75.98 ± 2.44 ^c^	1.46 ± 0.07 ^cd^	14.29 ± 0.33 ^cd^	6.84 ± 1.50 ^de^	23.32 ± 1.85 ^def^	7.07 ± 0.39 ^b^	5.70 ± 0.23 ^de^	49.91 ± 2.38 ^ab^	117.35 ± 2.43 ^c^
	75	1.45		71.94 ± 1.34 ^d^	2.30 ± 1.63 ^bc^	12.85 ± 1.46 ^ef^	9.93 ± 1.16 ^c^	34.10 ± 0.72 ^a^	5.84 ± 0.45 ^cde^	5.58 ± 0.48 ^e^	44.71 ± 0.79 ^cd^	115.46 ± 4.37 ^cd^

Control: freeze drying; HAD, hot air drying; IR-HAD, combined infrared and hot air drying; IR-HAD + HAD, sequential IR-HAD and HAD. Values with the different letters within each column indicate significant differences (*p* < 0.05).

**Table 3 foods-11-02240-t003:** Effects of drying methods on volatile compounds of chrysanthemum cakes.

No	Compounds	Formula	Content (μg/kg)			
			Control	HAD	IR-HAD	IR-HAD + HAD
	**Terpenes**					
1	1-Isopropyl-4-methylenebicyclo[3.1.0]hexane	C_10_H_16_	126.38	10.81	22.15	12.67
2	D-Limonene	C_10_H_16_	28.49	5.43	-	-
3	β-cis-Ocimene	C_10_H_16_	14.47	-	-	6.06
4	Ylangene	C_15_H_24_	-	-	9.15	5.19
5	trans-β-Bergamotene	C_15_H_24_	39.52	5.23	10.68	5.76
6	β-Caryophyllene	C_15_H_24_	144.34	40.17	45.29	45.52
7	(E)-β-Famesene	C_15_H_24_	227.11	21.21	37.44	26.74
8	β-Bisabolene	C_15_H_24_	686.14	189.02	287.07	232.92
9	γ-Muurolene	C_15_H_24_	164.78	124.97	321.08	190.58
10	α-Zingiberene	C_15_H_24_	391.72	294.86	632.19	545.52
11	Bisabolene	C_15_H_24_	99.51	43.45	67.44	64.84
12	(Z,E)-α-Farnesene	C_15_H_24_	178.91	89.74	166.21	136.33
13	α-Farnesene	C_15_H_24_	178.68	53.62	87.36	73.14
14	β-Sesquiphellandrene	C_15_H_24_	555.49	475.34	824.78	965.02
15	α-Curcumene	C_15_H_22_	172.97	101.50	153.41	131.91
16	Eucalyptol	C_10_H_18_O	330.23	16.20	28.33	20.16
17	(1R,2R,5S)-5-Isopropyl-2-methylbicyclo[3.1.0]hexan-2-ol	C_10_H_18_O	191.12	7.61	14.00	9.16
18	Linalool	C_10_H_18_O	108.97	6.27	8.11	7.77
19	Terpinen-4-ol	C_10_H_18_O	-	9.39	10.38	5.91
20	Isoborneol	C_10_H_18_O	1553.76	174.71	246.03	142.13
21	endo-Borneol	C_10_H_18_O	1618.74	-	-	-
22	Nerolidol	C_15_H_26_O	-	19.73	-	36.69
23	7-epi-γ-Eudesmol	C_15_H_26_O	70.26	8.61	58.67	34.52
24	Eudesm-7(11)-en-4-ol	C_15_H_26_O	158.83	122.90	100.63	250.93
25	Carveol	C_10_H_16_O	91.97	19.39	26.76	8.98
	**Ketones**					
26	6-Methyl-5-heptene-2-one	C_8_H_14_O	15.14	-	-	-
27	Bicyclo[3.1.1]hept-2-en-6-one, 2,7,7-trimethyl-	C_10_H_14_O	95.26	-	-	36.25
28	Bicyclo[2.2.1]heptan-2-one, 1,7,7-trimethyl-, (1S)-	C_10_H_16_O	1472.76	35.48	72.71	-
29	D-Carvone	C_10_H_14_O	101.51	11.90	16.50	11.29
30	l-Verbenone	C_10_H_14_O	128.40	3.02	-	5.11
	**Esters**					
31	trans-Chrysanthenyl acetate	C_12_H_18_O_2_	35.70	2.68	-	2.13
32	Bornyl acetate	C_12_H_18_O_2_	174.18	32.97	68.60	49.22
33	Methyl caprate	C_11_H_22_O_2_	88.94	11.06	14.10	20.13
34	Benzyl pentanoate	C_12_H_16_O_2_	265.17	38.33	49.94	26.08
35	Benzyl isovalerate	C_12_H_16_O_2_	46.80	7.45	13.00	10.07
	**Hydrocarbon**					
36	p-Xylene	C_8_H_10_	-	4.12	-	-
37	Undecane	C_11_H_24_	21.71	6.33	-	-
38	1,3,5,7-Cyclooctatetraene	C_8_H_8_	38.88	19.47	41.59	17.19
39	1-Pentadecene	C_15_H_30_	-	4.27	8.09	6.70
	**Aldehydes**					
40	Benzaldehyde	C_7_H_6_O	92.38	22.21	37.40	26.96
41	3-Hydroxybenzaldehyde	C_7_H_6_O	56.04	23.45	41.17	37.52
42	4-iPr-Benzaldehyde	C_10_H_12_O		4.72	6.68	
43	Furfural	C_5_H_4_O_2_	-	3.87	23.85	2.91
	**Alcohols**					
44	2,4-Dimethyl-4-octanol	C_10_H_22_O	-	19.15	-	5.66
45	3-Methyl-3-cyclohexen-1-ol	C_7_H_12_O	-	29.07	36.29	4.87
46	Benzyl alcohol	C_7_H_8_O	25.58	11.92	9.95	6.18
	**Phenols**					
47	Butylated Hydroxytoluene	C_15_H_24_O	-	5.97	11.02	-
	**Furans**					
48	Dihydro-β-agarofuran	C_15_H_26_O	57.27	13.62	14.37	15.60

Control: freeze drying; HAD, hot air drying at 65 °C; IR-HAD, combined infrared and hot air drying at 65 °C; IR-HAD + HAD, sequential IR-HAD and HAD at 65 °C. -: Not detected.

## Data Availability

The data presented in this study are available on request from the corresponding author.

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
