# Peer review of "Effect of Combined Infrared and Hot Air Drying Strategies on the Quality of Chrysanthemum (Chrysanthemum morifolium Ramat.) Cakes: Drying Behavior, Aroma Profiles and Phenolic Compounds"

_foods, 2022, doi:10.3390/foods11152240_

Round 1

Reviewer 1 Report

O manuscrito é interessante para a área de estudo e está bem estruturado e fundamentado. Sugiro que os autores acrescentem novos critérios de seleção de modelos, além dos usuais, como os critérios AIC e BIC.

Reviewer 2 Report

The research article entitled “Effect of Combined Infrared and Hot Air Drying Strategies on the Quality of Chrysanthemum (Chrysanthemum morifolium Ramat.) Cakes: Drying Behavior, Aroma Profiles and Phenolic Compounds” submitted in your esteemed journal is a good quality of work. The manuscript is well written and explained.

Here are some comments and suggestions given below: 

·         I don’t found the reference number 60 in the text.

·         Use proper Degree Celsius symbol during temperature representation e.g. in line no 127, 131, 155, table no.1 etc.

Overall the presentation of data and description of the results are well explained

“After reviewing the complete manuscript, in my opinion that this manuscript is accepted after minor revision for publication in your esteemed journal”.

Reviewer 3 Report

This study aimed to investigate the effects of different combined infrared and hot air drying strategies on the drying behavior, physicochemical properties and microstructure of medical chrysanthemum-based products. The idea is interest but there are some concern as below:

- Introduction: Please add some similar work has been done for other flowers or plants

line 130: what was vacuum condition for FD and time of drying?

line 247: " 1 g of dried" correct it.

Line 280: "100 mg of" correct it.

Line 282: change rpm to g

Statistical presentation in results should clarify such as Fig 5 how author anlaysed data and why data not analysed among temperature?

Reviewer 4 Report

line 127: indicate the reference of the applied method.

lines 242: indicate reference.

line 261: PCA - enter the full denomination and in parentheses PCA.

line 270: enter the full name and HCA in parentheses.

line 279: indicate the reference of the applied method.

line 284: indicate reference of the applied method

line 454: Chrysanthemum contains amino acids and reducing sugars for the Maillard reaction to occur?

line 465: this reference refers to carrots, not "chrysanthemum was rich in amino acids and reducing sugars". Review and change.

line 619: reference 50 is it talking about WBC and WHC? Corroborate.

line 646: reference [51] appears as [52] in references, review and following. 53, 54, 55 etc.

Round 2

Reviewer 3 Report

It can be accept but English language need to improve. For example starting of sentences should not with number 10g is not correct ten gram